# Fiber Bragg Grating Sensors for Pile Jacking Monitoring in Clay Soil

**DOI:** 10.3390/s20185239

**Published:** 2020-09-14

**Authors:** Yonghong Wang, Xueying Liu, Mingyi Zhang, Xiaoyu Bai, Ben Mou

**Affiliations:** 1School of Civil Engineering, Qingdao University of Technology, Qingdao 266033, China; hong7986@163.com (Y.W.); liuxueying@hnu.edu.cn (X.L.); zhangmingyi@qut.edu.cn (M.Z.); baixiaoyu@qut.edu.cn (X.B.); 2Shandong Province Higher Education School Blue Economic Zone Engineering Construction and Safety Collaborative Innovation Center, Qingdao University of Technology, Qingdao 266033, China

**Keywords:** fiber Bragg grating, pile jacking, model pipe piles, laboratory test, numerical analysis

## Abstract

The small deformation of the model pile in a pile jacking test makes its accurate measurement very difficult. Based on the installation method of clamping at both ends, a sensitized miniature fiber Bragg grating (FBG) strain sensor was developed to measure the pile strain of small-size model piles. This study investigated the working principle of the sensitized miniature FBG sensor and analyzed the strain transfer characteristics of the sensor by calibration test. The lateral resistance and the earth plug resistance of the model pile were measured accurately by the grooves embedded in the outer tube of the double-wall opening model pile and the sensors directly pasted on the surface of the inner tube. The results of this study show that the sensitized miniature FBG strain sensor has the advantages of high sensitivity, strong interference resistance, and high test accuracy. The FBG strain sensor was completely fixed on the pile by clamping supports at both ends, and the strain measured using the FBG strain sensor was found to be consistent with the pile deformation.

## 1. Introduction

In the process of pile jacking, the pile body is subjected to complex forces, in which the pile end breaking the soil faces the resistance of the pile end, and the outer wall of the pile body produces sliding friction with the soil, i.e., the lateral resistance of the pile and the wall of the open-pipe pile body exert resistance to the soil plug because of the earth plug effect, i.e., the resistance of the pile inside [1]. An indoor model test is the most direct method to study the mechanical properties of penetration of the piles. The problem of pile jacking has been investigated in detail through indoor model tests, in which most model piles are made of steel or aluminum pipes [2,3,4,5,6]. In order to avoid the boundary effect of the model test and satisfy the geometric similarity ratio of the model pile and the engineering pile, the diameter and length of the model pile are both kept small. The model piles in the centrifuge designed by Nicola et al. [7] and Klotz et al. [8] are all 16 mm in diameter, and the pile lengths are 350 and 375 mm, respectively. White et al. [9] designed a centrifugal model pile with a side length of 9 mm and a pile length of 185 mm. Zhang et al. [10] designed instrumented model piles with the inner and outer diameters of 12.7 and 16.0 mm, respectively, and a pile length of 600 mm. In the model test, the stress environment of the model pile during the pile jacking process is complicated, making the strain measurement of the model pile very difficult.

In the indoor model test, the strain of the model pile is measured using a traditional electric measuring strain gauge. Since the inside, outside, and end of the pile are in contact with the soil in the pile jacking process, the stress state of the soil around the pile end and pile body dynamically changes in the pile jacking process, and soil particle movement during the test will disturb the pile end and pile body; therefore, it is not suitable to attach the strain gauge on the outside surface of the model pile. In addition, the traditional test sensor exhibits poor anti-electromagnetic interference ability and is greatly affected by environmental changes such as external temperature, causing a large error; thus, the measured value is significantly different from the actual value, affecting measurement accuracy. At the same time, the small size of the model pile in the model test results in a small strain and thus, has stringent requirements on the size and sensitivity of sensors. Traditional electric sensors cannot meet these special test requirements [11,12]. In the geotechnical centrifugal model test, fiber Bragg grating (FBG) sensing technology was used to monitor the strain of the pile body caused by the collapse of collapsible loess around piles in loess areas [13]. However, the pile strain caused by the collapse of the soil around the pile cannot reflect the pile stress in the pile jacking process. Especially for the inner and outer pipes of the open pipe pile, the main goal of strain sensor application in the indoor model test is to accurately measure the strain change of the model pile and reasonably install the sensor without affecting the pile jacking process.

A fiber Bragg grating (FBG) sensor has strong anti-electromagnetic interference ability and can realize temperature self-compensation; therefore, it is less affected by external temperature and other environmental changes. At the same time, this type of sensor is small in size and light in mass, and multiple gratings can be written in one fiber. In recent years, FBG sensors have been widely used in various practical applications [14,15,16]. FBG sensing technology can monitor pile deformation in the process of pile foundation static load and drawing tests. FBG sensors have been embedded in the pile to reveal the stress mechanism of the pile [17,18,19]. For the strain test of the pile body in the pile jacking process, it is very important to overcome the serious disturbance caused by soil particle movement to the pile end and pile body and to protect the sensor from damage during the test. Many researchers proposed the measures of embedding and encapsulating FBG sensors in the grooving of the pile body [20,21,22]. The test results show that the FBG sensor exhibited reliable data in monitoring pile jacking and is suitable for axial strain measurement of a pile. These research results are mainly based on prestressed high-strengh concrete (PHC) pipe pile diameter more than 300 mm and pile concrete elastic modulus of 36 GPa in a field test. However, in the laboratory test with small model pile size, the elastic modulus of steel or aluminum used to make the model pile is 70 and 206 GPa, respectively, resulting in a small deformation in the pile jacking process, which reduces the test accuracy of the FBG sensor.

In this study, a sensitized micro-FBG strain sensor was proposed and applied to the pile jacking test of indoor model piles. In view of the small size of the indoor model pile and the limited strain range of the pile, the FBG strain sensor was installed by the clamping method. In order to explore the applicability of the sensitized miniature FBG strain sensor in the pile jacking model test, the pile jacking test was carried out by changing the pile diameter, pile length, and pile end form. The results show that the strain sensor can effectively monitor the strain of the indoor model piles.

## 2. Operational Principles

### 2.1. Working Principle of the FBG Sensor

FBG is a type of fiber core refractive index that changes periodically and forms a Bragg grating with spatial phase in the fiber core by ultraviolet holographic exposure with germanium, phosphorus, and other fiber materials’ nonlinear absorption effect. As shown in Figure 1, light with a specific center wavelength is reflected at the grating and is represented by the Bragg wavelength *λ_B_* (nm). The spectral width of the grating period Λ is in the range 0.05–0.3 nm, and all other ordinary light passes through the optical fiber grating [23]. The Bragg wavelength *λ_B_* satisfies Bragg conditions [24]:(1)$$\lambda_{B}=2n_{eff}\Lambda$$
where $n_{eff}$ is the effective refractive index of the optical fiber core.

The fiber grating period and effective refractive index will be affected by the strain and temperature, and at the same time, the photoelastic effect can also change the grating refractive index, which, in turn, changes the FBG wavelength. The change in *λ_B_*, namely Δ*λ_B_*, can be measured by Equation (1). In this study, the FBG strain sensor adopts a two-end clamping package. When the temperature increases, the wavelength shift of the internal grating can be offset by the negative expansion material, and thus, the response to the temperature change can be ignored and Δ*λ_B_* can be expressed by Equation (2) [26]:(2)$$\frac{\Delta \lambda_{B}}{\lambda_{B}}=\left\{1-\frac{n_{eff}{}^{2}}{2}\left[p_{12}-\upsilon (p_{11}+p_{12})\right]\right\}\varepsilon =\left\{1+\frac{n_{eff}{}^{2}}{2}\left[\upsilon (p_{11}+p_{12})-p_{12}\right]\right\}\varepsilon$$
where $p_{11}$ and $p_{12}$ are the photoelastic effect constants; υ is Poisson’s ratio; ε is the axial strain of optical fiber.

Set $K_{\varepsilon }=\left\{1-\frac{n_{eff}^{2}}{2}\left[p_{12}-\upsilon \left(p_{11}+p_{12}\right)\right]\right\}$ to:(3)$$\Delta \lambda_{B}=K_{\varepsilon }\cdot \lambda_{B}\cdot \varepsilon$$

### 2.2. Measurement of Jacked Pile Deformation

Fiber at both ends of the fiber grating is firmly embedded in the clamping support, and both ends of the clamping support are fixed. The deformation of the clamping FBG sensor is similar to that of the end bearing pile; therefore, the axial deformation of the clamping support is the strain of the FBG, and the wavelength change of the FBG can be calculated according to Equation (2). The clamping support is fixed on the pile body, and its bottom is leveled to make the clamping support and the pile body in the same level. Therefore, the deformation of the clamping support is consistent with the deformation of the pile, and thus, the axial strain of the clamping support can accurately reflect the deformation of the pile.

The axial strain of the clamping support is calculated based on the theory of material mechanics. Figure 2 shows the arrangement of clamping support and FBG. Assuming that the test member undergoes an axial deformation ΔL between the two clamping supports, the corresponding clamping support and fiber grating deformation are $\Delta L_{s}$ and $\Delta L_{f}$, respectively. Assuming no deformation in the fiber and the clamping support leads to the following equations [27]:(4)$$\Delta L_{S}=\frac{P_{S}L_{S}}{E_{S}A_{S}}$$
(5)$$\Delta L_{f}=\frac{P_{f}L_{f}}{E_{f}A_{f}}$$
where $E_{s}$ and $E_{f}$ are the elastic modulus (N/mm^2^) of the clamping support and FBG, respectively. $A_{s}$ and $A_{f}$ are the cross-sectional area (m^2^) of the clamping support and the fiber Bragg grating, respectively. $P_{s}$ and $P_{f}$ are the internal forces (N) generated by the clamping support and FBG, respectively.

The internal force generated by the clamping support and the FBG is evenly distributed, and the axial strain of the clamping support and the FBG can be expressed as:(6)$$\frac{\varepsilon_{s}}{\varepsilon_{f}}=\frac{\frac{\Delta L_{S}}{L_{S}}}{\frac{\Delta L_{f}}{L_{f}}}=\frac{E_{f}E_{f}}{E_{s}A_{s}}$$

The clamping support is made of steel tube. The diameter of the clamping support and FBG grating are 0.8 and 0.125 mm, respectively. Substituting the clamping support $E_{s}=2.1\times 10^{8}kPa$ and fiber Bragg grating $E_{f}=7.2\times 10^{7}kPa$ into Equation (6) leads to the following equation:(7)$$\frac{\varepsilon_{s}}{\varepsilon_{f}}=0.0084$$

According to Equation (8), the deformation of the clamping support is smaller than that of the FBG, and thus, can be ignored in the sensor design. When the sensor is subjected to axial force, the axial strain of the supporting supports at both ends is the same as that of the FBG. When the center wavelength of the optical fiber core in the sensor made of fused silica is in the 1550 nm band, $K_{\varepsilon }\approx 1.2pm/\mu \varepsilon$, combined with Equation (2), the relationship between the deformation of the sensor and the deformation of the test member can be expressed as:(8)$$\varepsilon =\frac{L_{f}}{L}\varepsilon_{f}=\frac{L_{f}\Delta \lambda_{B}}{1.2L}$$

Equation (8) shows that by adjusting the ratio of $L_{f}$ to L of the fiber grating strain sensor, the sensitivity coefficient of the sensor can be changed.

## 3. Sensor Calibration

The fiber core of the sensitized micro FBG strain sensor adopts naked fiber grating; the fiber coating layer adopts polyamide and the fiber Bragg grating is sealed with binder. The calibration experiment of the sensitized miniature FBG strain sensor was realized by calibrating the bare fiber grating, as shown in Figure 3. The FBG strain sensor and bare FBG are installed on both sides of the elastic beam and loaded using a universal test machine (MTS Criterion 40). The calibration test principle diagram is shown in Figure 4. In order to ensure that the strain range of FBG strain sensor and bare fiber grating is in line and the strain value is the same, the load value was increased from 0 to 1000 με. The calibration experiment was repeated and the average value of the results was obtained.

From the calibration test data, the relationship curve between the Bragg wavelength of the calibrated FBG strain sensor and the wavelength of the bare fiber grating can be drawn. Figure 5 shows the calibration test results, indicating that the FBG strain sensor has a good linear fitting degree with bare fiber grating, with a linear correlation coefficient greater than 0.9999. The strain sensitivity of bare fiber gratings is 1.2 pm/με. According to the calibration results in Figure 5, the sensitivity coefficient of the FBG strain sensor is 2.004 × 1.2 = 2.4048 pm/με.

Repeated calibration experiments were carried out using a universal experimental machine (MTS Criterion 40) equipped with an FBG strain sensor, a calibration steel plate (length × width × height = 400 × 100 × 5 mm), and a fiber grating demodulator, and finally, the average value was taken, as shown in Figure 3. The FBG strain sensor and bare fiber grating are installed on both sides of the calibrated steel plate, and the load value of the calibrated steel plate was increased from 0 to 1000 με, to ensure that the strain online elastic range of the FBG strain sensor and bare fiber grating is within the same strain value.

## 4. Model Pile, Sensor Installment, and Instrumentation Setup

### 4.1. Model Pile

This study is based on the self-developed double-wall opening model pipe pile, with an effective total length of 1065 mm. The model pipe pile comprises two concentric thin-walled circular pipes with a thickness of 3 mm, an outer diameter of 140 mm, and an inner diameter of 120 mm. The round tubes are made of aluminum, with an elastic modulus of 72 GPa and Poisson’s ratio of 0.3. There are symmetrical grooves on both sides of the surface of the outer pipe pile (depth × width = 2 × 2 mm), and two holes with a diameter of 5 mm are arranged on one side of the pile to encapsulate the FBG sensor and the incoming line. The outer wall of the inner pipe is directly pasted with the FBG sensor. The inner and outer pipes of the double-wall opening model with pile caps are shown in Figure 6.

### 4.2. Finite Element Simulation

The large finite element numerical software Abaqus was used for simulation analysis. Solid elements were used to simulate the pile and soil. The soil layer was 1.1 m wide and 2.2 m deep. The body boundary is fixed for the side and the lower surface, and the ground surface is free. Considering that the process of static pressure pile penetration is actually the process of pile–soil interface squeezing and sliding, the surface–surface contact model is adopted here. The friction type of the contact surface is Coulomb friction, and it is assumed that the pile–soil contact will not be separated later, that is, the pile–soil is in contact and sliding state. The numerical calculation takes the end of piling as the termination condition.

#### 4.2.1. Material Constitutive Structure

The pile is made of aluminum alloy, its mass density is 2.7 × 10^−3^ g/mm^3^, elastic modulus is 72 GPa, Poisson’s ratio is 0.3, and the pipeline constitutive model adopts the linear elastic model. The modified Cam-Clay constitutive was used for soil calculation, and the Cambridge model in Abaqus finite element calculation needs to be combined with the porous elastic model.

#### 4.2.2. Finite Element Results

The same amount of compressive stress was applied to the upper and lower parts of the model pipe pile. The Abaqus simulated stress distribution diagram of the FBG strain sensor on the inner and outer pipes is shown in Figure 7. It can be seen from Figure 7 that stress concentration phenomenon occurs at the pile head of the inner and outer pipes under compressive stress. According to Abaqus strain distribution (Figure 7a,b), for deformation under compressive stress, the pasted stress and strain distribution on the inner tube of the FBG strain sensor is uniform, and the grooving of the outer tube of the FBG strain sensor has little influence on the stress and strain. By observing the strain distribution cloud map of Abaqus, it can be seen that the slotting has almost no effect on the stress and strain of the cylinder. It can be seen from Figure 7c that the stress around the opening varies greatly, and the stress around the opening is distributed like “petals”. In the figure, the stress of the left and right four pieces of “petals” is larger, while the stress of the upper and lower two pieces is smaller. According to the numerical calculation results, in the double-wall opening model pile, miniature FBG strain sensors with sensitization were installed at 20 mm from the outer pipe to the pile end and 80 mm from the pile top to measure the changes in the pile end resistance and pile pressure during pile jacking process. The FBG strain sensors of the inner pipe were placed at 20 mm from the pile end to monitor the earth plug resistance of the inner pipe.

### 4.3. Test Instruments

In order to ensure the survival rate of the sensor and its easy installation, the sensor in the outer pipe is arranged in the way of packaging after surface grooving. First, the clamping sleeve of one end of the sensor was attached to the bottom of the grooving with 502 glue, and then, the other end was pre-stretched to the outside in the range 0.2–0.8 nm before being attached. Finally, the sensor was packaged with epoxy resin to protect the sensor from environmental interference. The FBG strain sensor in the inner pipe is directly attached to the surface of the inner tube pile body and coated with 702 silica gel for protection. The installation details of the FBG sensors in the outer and inner pipes are shown in Figure 8. The FBG data acquisition instrument adopts the FS2200RM-Rack-Mountable Bragg Meter demodulation instrument produced in Portugal, which can realize 6-channel simultaneous dynamic acquisition with an acquisition frequency of 1 Hz.

### 4.4. Experimental Procedure

In order to explore the applicability of FBG sensing technology in the jacked pile jacking resistance test in cohesive soil, a series of model pipe pile jacking tests were conducted in homogeneous cohesive soil. At the same time, the penetration characteristics of the pile end resistance and pile side resistance during the pile jacking process were analyzed. The pile jacking test device includes a data acquisition system, a loading system, and a model box system. The whole pile jacking process is completed in two steps, with a pause in the middle to increase the drop height of the jack. The pile jacking speed is approximately 300 mm/min. The jacked pile experiment was carried by changing the pile diameter, pile length, and pile end form parameters, keeping all other parameters fixed. The inner and outer pile side resistances were isolated through the sensitization of the miniature FBG strain sensor on the top and end of the pile body strain measurement directly, and the process of pile jacking pile side resistance and pile end resistance characteristics was analyzed to investigate the change in pile jacking resistance.

## 5. Results and Discussion

According to the mechanism of jacked pile, the pile resistance *R* is composed of pile end resistance *Q*_s_ and pile side resistance *F*_s_. The pile end resistance is approximately based on the axial force measured by the FBG sensor nearest to the pile end, namely:*R* = *Q*_s_ + *F*_s_(9)

By recording the initial wavelength of the FBG sensor and the change of the wavelength in the penetration process, the change of the wavelength is converted into the pile strain in the penetration process. The strain of pile body is multiplied by the elastic modulus of pile body to calculate the stress change of pile body and the axial force change of pile body is calculated from the stress change of pile body, as shown in Equations (10) and (11):(10)$$\Delta \varepsilon =\frac{\left(\lambda_{1}-\lambda_{0}\right)}{K}\times 1000$$
where, Δε is the microstrain value; $\lambda_{0}$ is the initial central wavelength value (nm) of the fiber grating; $\lambda_{1}$ is the central wavelength value (nm) of the fiber grating during penetration. K is strain sensitivity coefficient (pm/με).
(11)N=E⋅Δε⋅A
where, *N* is the pile axial force (kN) at the position of FBG sensor; *E* is the elastic modulus of the model pile (MPa); *A* is the cross-sectional area of the pile (mm^2^).

### 5.1. Influence of Pile Diameters on Pile End Resistance and Pile Side Resistance

Figure 9 shows the effect of different pile diameters on the pile end resistance and pile side resistance, indicating that both types of resistance are closely related to pile diameter. The larger the pile diameter, the greater the pile end resistance and pile side resistance. Compared to the pile end resistance, the trend and value of pile side resistance with increasing depth are smaller than the pile end resistance, indicating that the penetration resistance in the process of pile jacking is mainly pile end resistance. When the pile jacking test is carried out in viscous soil, the penetration resistance mainly comes from the pile end resistance generated by punching shear when the pile end passes through the soil and is greater than the pile side resistance generated in the process of pile jacking. Under the same pile jacking depth, the pile side resistance of TP4 is less than that of test pile TP2; therefore, it can be concluded that the pile side resistance is significantly affected by the pile diameter.

### 5.2. Effect of Pile Length on Pile End Resistance and Pile Side Resistance

Figure 10 shows the effect of pile length on the pile end resistance and pile side resistance, indicating that pile length has a slight effect on the pile end resistance and pile side resistance; the change rules of pile end resistance and pile side resistance of test piles TP2 and TP3 are basically the same and the difference between the pile end resistance and pile side resistance within the same range of pile length is not much. When the pile depth of TP3 increased from 90 to 110 cm, the pile end resistance increased from 1.812 to 2.054 kN and the pile side resistance increased from 0.939 to 1.244 kN and is similar to the pile side resistance of TP2. The test result of the pile with the length of 110 cm is slightly larger than that with the length of 90 cm, and is related to the buried depth of the test pile TP3. The pile side resistance gradually increases with increasing pile jacking depth. At the same depth, the pile side resistance of TP3 is smaller than that of the test pile TP2. The pile side resistance of the two test piles is less than 0.1 kN within the range of the buried depth of 10 cm. The main reason is that during the pile jacking process, the pile body moves in the shallow soil, creating a certain gap between the pile and soil, and the contact is not close. At the end of pile jacking, the maximum pile side resistance of TP3 is about 4.5% higher than that of TP2.

### 5.3. Effect of Pile End Form on Pile End Resistance and Pile Side Resistance

Figure 11 shows the effect of pile end form on the pile end resistance and pile side resistance, indicating that both of them are related to the pile end form, and the side resistance of the closed-end pile is obviously greater than that of the open-end pile. At the beginning of the pile jacking stage, the pile end resistance of TP1 is less than that of TP2. With increasing pile jacking depth, the pile end resistance of TP1 and TP2 gradually approaches, i.e., the soil plug of TP1 is formed, and the soil plug resistance gradually develops until the height of the soil plug no longer changes. The pile side resistance is affected by the pile end form. At the same burial depth, the pile side resistance of the open-end test pile TP1 is less than that of the closed-end test pile TP2.

### 5.4. Percentage of Pile End Resistance and Pile Side Resistance in Pile Jacking Resistance

The above analysis indicates that the pile jacking resistance in the homogeneous viscous soil layer is mainly the pile end resistance and the pile side resistance is relatively small, as shown in Table 1. At the same time, the pile diameter and pile end form significantly affect the pile jacking resistance in the homogeneous viscous soil layer, but the pile length has no effect on the pile jacking resistance.

### 5.5. Comparison of Pile Jacking Resistance Curve

Figure 12 shows the relationship curve between the resistance of model piles and the burial depth, indicating that for the open pile TP1 and closed pile TP2 with equal pile diameter and pile length, the pile jacking resistance rises rapidly in the initial stage. However, when the burial depth is about 20 cm, the pile jacking resistance of TP1 and TP2 is equal, and the pile jacking resistance shows an inflection point, followed by a slow increase. When the burial depth is about 40 cm, the pile jacking resistance of the open pile TP1 starts to decrease more than that of closed pile TP2, and the resistance growth rate of TP2 is higher than that of TP1. When the burial depth is 90 cm, the pile jacking resistance of TP2 is 15.8% higher than that of TP1, and this observation is consistent with the mechanism of the penetration process of the open and closed piles. The larger the diameter of the pile, the faster the increase rate of the pile jacking resistance of the closed pile. For TP2 and TP4 of equal length of piles, at a burial depth of 90 cm, the pile jacking resistance of TP2 is 31.3% higher than that of TP4. For TP2 and TP3 with equal diameters of piles, when the burial depth is in the range 0–90 cm, the pile jacking resistance of the two piles is basically equal. When the burial depth is 110 cm, the pile jacking resistance of TP2 increases by 12.3% compared to that of TP3. To sum up, the pile bearing capacity can be increased by closing the pile end, increasing the pile diameter, and increasing the pile length. The increase in pile diameter leads to the largest increase in the pile bearing capacity. The effect of the pile end form is less, and the increase in the pile length is the smallest.

### 5.6. Relationship between Pile Top Load and Pile End Resistance and Pile Side Resistance

Figure 13 shows the relationship between the pile top load and pile end resistance and pile side resistance, indicating that the pile outside resistance of TP1 is greater than that of the pile inside resistance. When the burial depth is small, the change in the pile inside resistance is obvious. With increasing burial depth, the pile inside resistance increases slowly and basically remains unchanged. During the pile jacking process, both the pile outside and inside resistance maintain an increasing trend. The sum of the pile inside resistance, pile outside resistance, and pile end resistance of TP1 is equal to the pile top load, and the sum of the pile side resistance and pile end resistance of TP2, TP3, and TP4 is equal to the pile top load, fully demonstrating the effectiveness of the test results.

## 6. Conclusions

A sensitized miniature FBG strain sensor was introduced, calibrated, and further applied to a pile jacking test of double-wall open model pipe piles. The strain of the small size model piles was monitored. The clamping installation method at both ends was designed to not only enhance the strain sensitivity, but also ensure consistent deformation of the sensor and the pile body. Based on the experimental results of this study, the following conclusions can be summarized:(1)The sensor can better measure the strain of the pile by enhancing the strain sensitivity.(2)The encapsulation method of the pile sensor proposed in this study effectively prevented the damage to the FBG sensor during the pile jacking process in clay soil.(3)The sensor met all the test requirements of pipe pile jacking resistance in cohesive soil and clearly reflects the change law of pile end resistance and pile side resistance under different pile diameters, pile lengths, and pile end forms.(4)The pile jacking resistance in the homogeneous cohesive soil layer is mainly the pile end resistance and the pile side resistance is small during the pile jacking process. The pile diameter and the pile end form significantly affect the pile jacking resistance in homogeneous cohesive soil layers, and the pile length has a smaller effect on the pile jacking resistance.

In summary, the sensitized miniature FBG strain sensor can accurately measure the strain of the small-size model pile and effectively monitor the pile outside and inside resistance of the double-wall open model pipe pile.

## Figures and Tables

**Figure 1 sensors-20-05239-f001:**
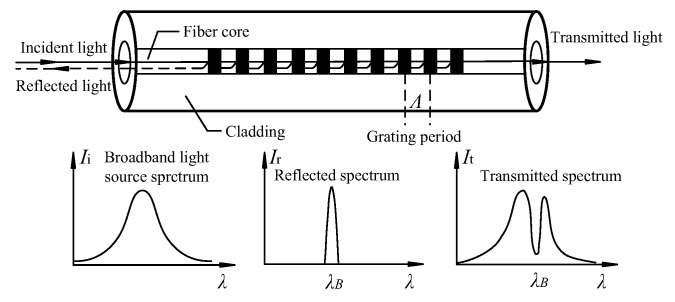
Functioning principle of FBG sensing technology (adapted from [25]).

**Figure 2 sensors-20-05239-f002:**
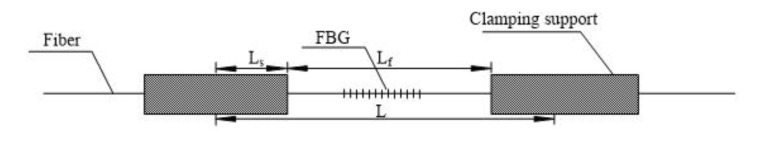
Deflection of clamping support.

**Figure 3 sensors-20-05239-f003:**
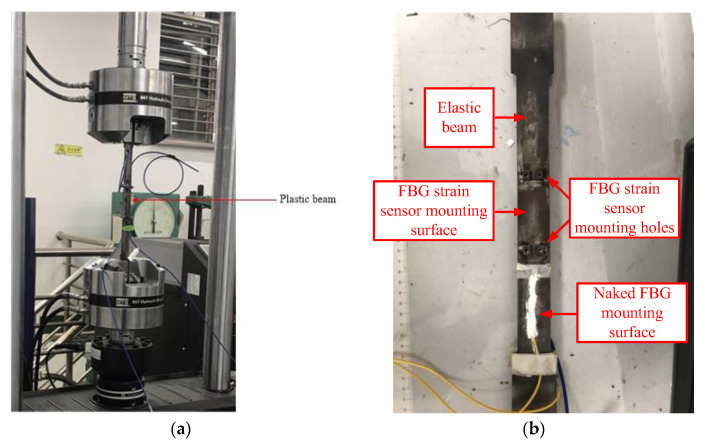
The calibration tests using the universal material test system: (**a**) Overall schematic diagram, (**b**) Detail view of elastic beam (adapted from [28]).

**Figure 4 sensors-20-05239-f004:**
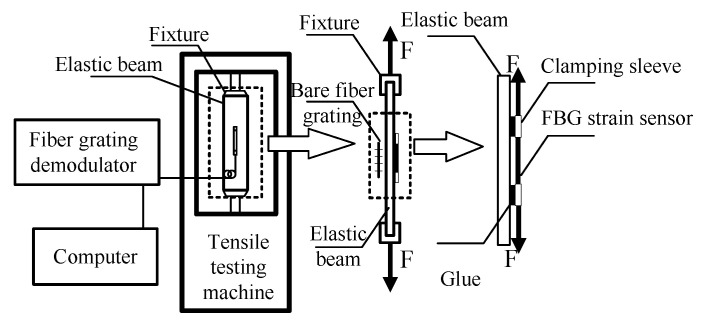
Calibration schematic of FBG strain sensors.

**Figure 5 sensors-20-05239-f005:**
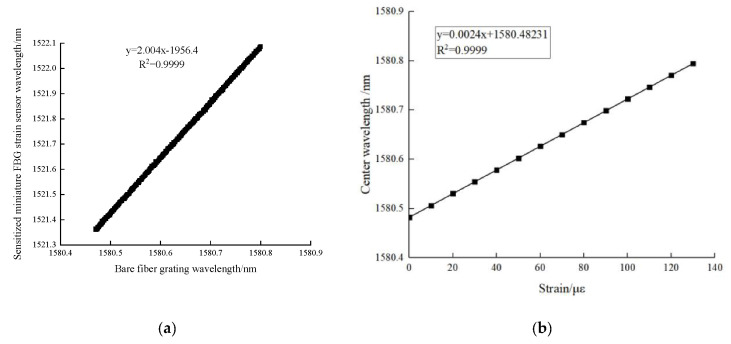
Results of FBG strain sensor calibration tests: (**a**) The relation curve between Bragg wavelength and BFBG wavelength, (**b**) The relation curve between wavelength and strain of bare fiber Bragg grating.

**Figure 6 sensors-20-05239-f006:**
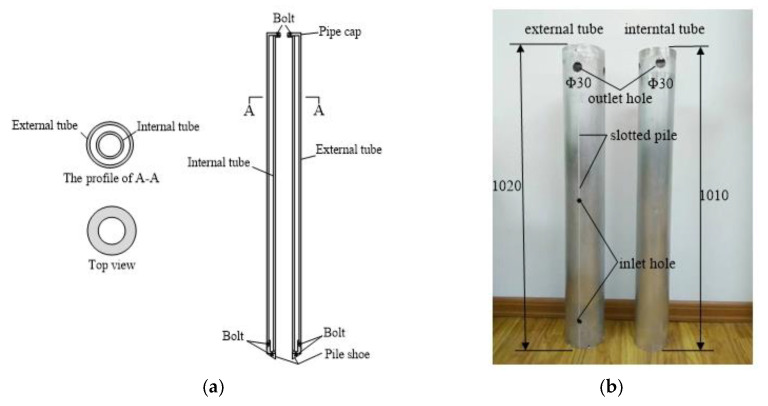
Internal and external tube of the double-walled open mode pipe pile: (**a**) Schematic diagram; (**b**) Physical map (unit: mm).

**Figure 7 sensors-20-05239-f007:**
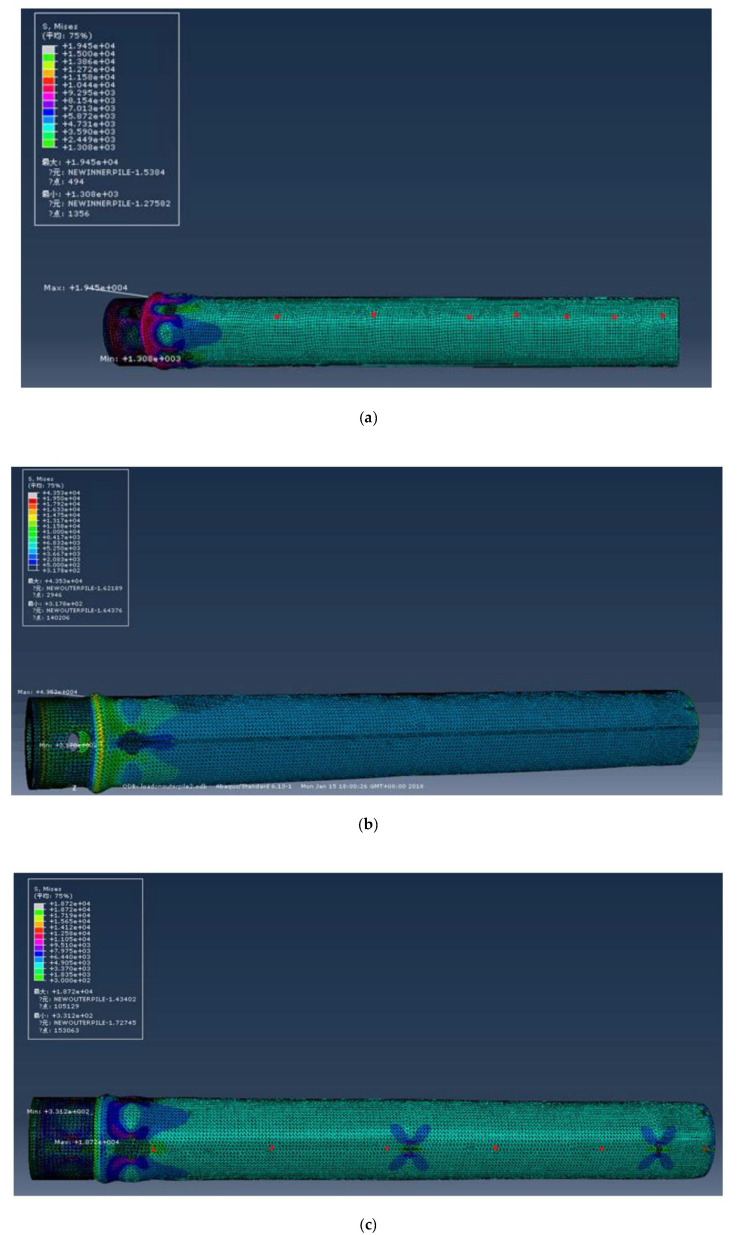
The simulated stress distribution of FBG strain sensor through Abaqus: (**a**) Internal pipe; (**b**) Slotted on external pipe; (**c**) Opening on outer pipe.

**Figure 8 sensors-20-05239-f008:**
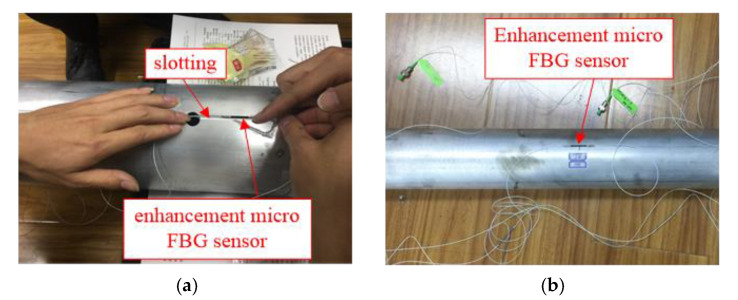
FBG sensor installation details of (**a**) outer and (**b**) inner pipe.

**Figure 9 sensors-20-05239-f009:**
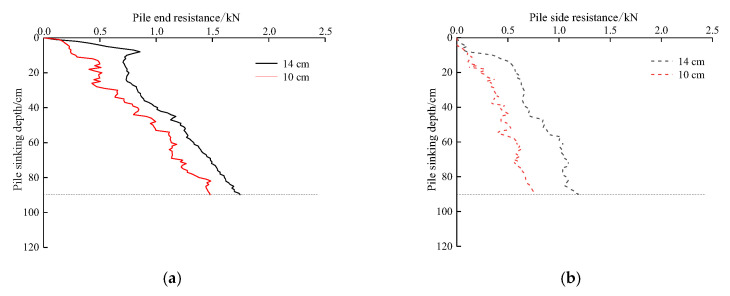
Relationship between the pile jacking resistance and pile diameters: (**a**) Pile end resistance; (**b**) Pile side resistance.

**Figure 10 sensors-20-05239-f010:**
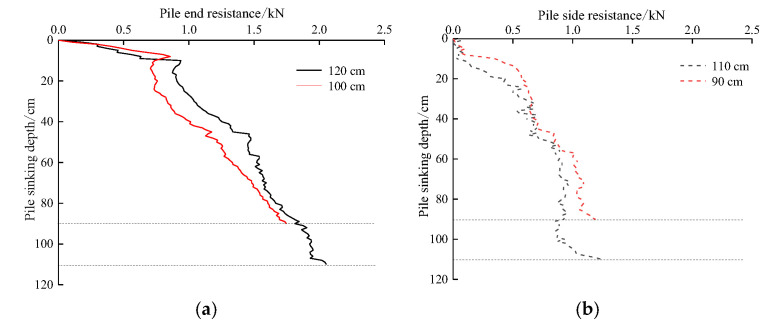
Relationship between pile jacking resistance and pile length: (**a**) Pile end resistance; (**b**) Pile side resistance.

**Figure 11 sensors-20-05239-f011:**
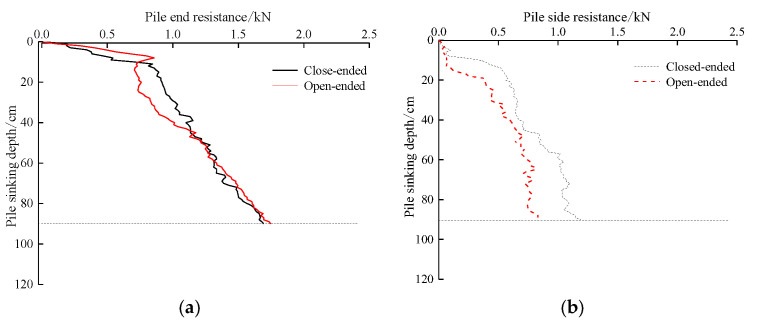
Relationship between pile jacking resistance and pile end form: (**a**) Pile end resistance; (**b**) Pile side resistance.

**Figure 12 sensors-20-05239-f012:**
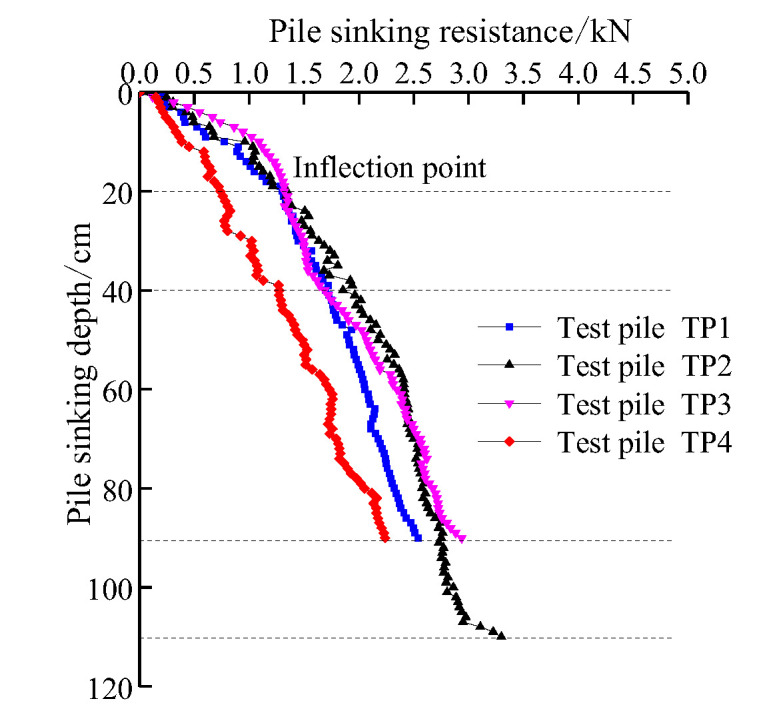
Comparison between pile sinking resistance curves of pile penetration.

**Figure 13 sensors-20-05239-f013:**
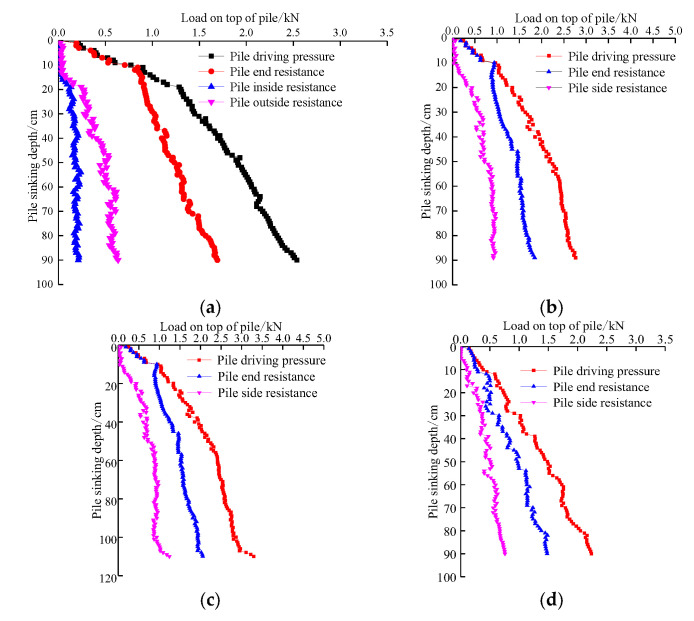
The relationship between the applied pile top load and pile end resistance and pile side resistance: (**a**) TP1; (**b**) TP2; (**c**) TP3; (**d**) TP4.

**Table 1 sensors-20-05239-t001:** Percentage of pile end resistance and pile side resistance at the end of pile jacking.

Test Pile Number	Pile Jacking Resistance (kN)	Pile End Resistance (kN) (Percentage)	Pile Side Resistance (kN) (Percentage)
TP1	2.538	1.692 (66.7)	0.846 (33.3)
TP2	3.298	2.054 (62.3)	1.244 (37.7)
TP3	2.938	1.747 (59.5)	1.191 (40.5)
TP4	2.238	1.480 (66.2)	0.757 (33.8)

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
