# Peer review of "Fiber Bragg Grating Sensors for Pile Jacking Monitoring in Clay Soil"

_sensors, 2020, doi:10.3390/s20185239_

Round 1

Reviewer 1 Report

This paper used FBGs for pile jacking monitoring in clay soil using FBG in a scaled lab experiment was the forces are much lower for conventional stain gages. This paper is reasonably well written although there are a few points which need to be clarified.  

  • I believe Fig 1, 3 seem to be taken from other sources (e.g. books) and if so should be referenced or redrawn.

Minor points:

  • Line 70 what is “micro FBG” this is used also in several places in the text? Most fibers are 125 μ Did the authors use other type of fiber? eg 80 μm outer diameter to make it smaller?
  • Line 135 “core in the sensor made of pure quartz “ ? Isn’t the fiber from fused silica?
  • Figure 3 is not particularly informative a comment or diagram could help.
  • Line 143 to 145 “The fiber core of the sensitized micro FBG strain sensor adopts pristine FBG; the fiber Bragg grating coating layer adopts polyamide, and the encapsulation material coating layer adopts fiber reinforced Polymer (FRP), eliminating the effect of binder on the strain signal transmission of FBG “ don’t quite make senses and need to be we-written. What is a “pristine FBG”  and how it eliminates “the effect of binder on the strain signal transmission of FBG sensor”
  • In lines 146 to 148 “The FBG strain sensor and pristine FBG are installed on both sides of the elastic beam and loaded using a universal test machine (MTS Criterion 40).” The authors seam to use two FBGs , but they don’t seem to mention this in any procedure. A schematic layout and procurers would be useful to explain all aspects of the testing layout.
  • Line 197 what is “502 glue”
  • Line 198 “pre-stretched to the outside in the range 0.2–0.8 nm before”? What do the authors mean here? nm is not strain units, are they referring to the FBG wavelength?  

All the result shown in figures 7 to 12 show a “point of inflection” Perhaps an explanation can be given for this behavior?

Reviewer 2 Report

The paper subject is related to development of FBG sensor for application of pile jacking monitoring.

The paper subject is very interesting and important from the practical point of view.

Unfortunately, the number of shortcomings in the presented study does not allow the paper to be published in the presented form.

Remarks:

1) “The FBG strain sensor adopted in this study is a negative expansion material introduced by packaging.” How the theoretical idea is realised in practice. Which material was applied and what is an accuracy of the proposed compensation solution.  

2) There are mistakes in the equations. Please, consider verification units in the equations, e.g. the sensitivity coefficient.

3) Sensor calibration. In a purpose of confirming values of the sensitivity coefficient understood as 1.2 pm/microepsilon a figure presenting relationship between the central wavelength change and strain being an effect of tensile force is required. Figure 4 presents a relationship between two sensors and does not allow to determine the sensitivity coefficient value.

4) Numerical model. There is lack of information about the model assumptions, type of elements etc. Additionally, the presented results are completely unreadable.

5) Results and discussion. All results are presented as relationships between force [kN] and depth [cm]. It can be guess that the section presents experimental results. Unfortunately, as the theoretical introduction is concerned on strain due to mechanical force, strain values measured by proposed FBG senor will be expected. If the authors would like to provide force values instead of strain, they should present the calculations (assumptions) related to determining force from the FBG sensor readings. As it was highlighted in the introduction, the pile jacking is a complex problem from the mechanical point of view.

Round 2

Reviewer 2 Report

The paper was improved. However, there are shortcomings that have to be corrected prior publication.

Remarks:
1) The packaging should be better explained. What about shear stress/strain on boundaries between outer tube and inner tube as well as between the inner tube and the fibre? Better explanation of equation 2 is required as I have not found it in the text of ref [26].
2) There are still mistakes in the equations.
3) The ABAQUS calculation results are still unreadable.

Author Response

Sorry for the late return, please understand. Please see the attachment.
